# The Risk of Breast Cancer According to Mutation Type and Position in Carriers of a Pathogenic Variant in *BRCA1*

**DOI:** 10.3390/curroncol32120705

**Published:** 2025-12-15

**Authors:** Joanne Kotsopoulos, Adriana I. Apostol, Kelly Metcalfe, Dimitri Jorgji, Cezary Cybulski, Jacek Gronwald, Jan Lubinski, Pal Moller, Raymond H. Kim, Amber Aeilts, Teresa Ramón y Cajal, Tuya Pal, Louise Bordeleau, Beth Y. Karlan, Christian F. Singer, William D. Foulkes, Fergus J. Couch, Dana Zakalik, Robert Fruscio, Nadine Tung, Ping Sun, Alvaro N. Monteiro, Steven A. Narod, Mohammad R. Akbari

**Affiliations:** 1Women’s College Research Institute, University of Toronto, Toronto, ON M5S 1A1, Canada; joanne.kotsopoulos@wchospital.ca (J.K.); adrianaionelia.apostol01@icatt.it (A.I.A.); kelly.metcalfe@utoronto.ca (K.M.); dimitri.jorgji@uwaterloo.ca (D.J.); ping.sun@wchospital.ca (P.S.); mohammad.akbari@utoronto.ca (M.R.A.); 2Dalla Lana School of Public Health, University of Toronto, Toronto, ON M5S 1A1, Canada; 3Dipartimento Scienze della Salute della Donna, Del Bambino e di Sanità Pubblica, Fondazione Policlinico Universitario Agostino Gemelli, IRCCS, Universita’ Cattolica del Sacro Cuore, 00168 Rome, Italy; 4Bloomberg Faculty of Nursing, University of Toronto, Toronto, ON M5S 1A1, Canada; 5Department of Genetics and Pathology, International Hereditary Cancer Center, Pomeranian Medical University, 70-204 Szczecin, Poland; cezary.cybulski@pum.edu.pl (C.C.); jacek.gronwald@pum.edu.pl (J.G.); jan.lubinski@pum.edu.pl (J.L.); 6Institute of Cancer Research, Department of Tumour Biology, The Norwegian Radium Hospital, Oslo University Hospital, 0379 Oslo, Norway; moller.pal@gmail.com; 7Princess Margaret Cancer Centre, University Health Network, Toronto, ON M5G 2C4, Canada; raymond.kim@uhn.ca; 8Division of Human Genetics, The Ohio State University Medical Center, Comprehensive Cancer Center, Columbus, OH 43210, USA; amber.aeilts@osumc.edu; 9Department of Medical Oncology, Hospital de la Santa Creu i Sant Pau, 08025 Barcelona, Spain; tramon@santpau.cat; 10Vanderbilt-Ingram Cancer Center, Department of Medicine, Vanderbilt University Medical Center, Nashville, TN 37232, USA; tuya.pal@vumc.org; 11Department of Oncology, Juravinski Cancer Centre and McMaster University, Hamilton, ON L8V 5C2, Canada; bordeleaul@hhsc.ca; 12Department of Obstetrics and Gynecology, David Geffen School of Medicine, University of California, Los Angeles, CA 90095, USA; bkarlan@mednet.ucla.edu; 13Department of Obstetrics and Gynecology and Comprehensive Cancer Center, Medical University of Vienna, 1090 Vienna, Austria; christian.singer@meduniwien.ac.at; 14McGill Program in Cancer Genetics, Department of Oncology, McGill University, Montreal, QC H3A 0G4, Canada; william.foulkes@mcgill.ca; 15Division of Experimental Pathology and Laboratory Medicine, Department of Laboratory Medicine and Pathology, Mayo Clinic, Rochester, MN 55905, USA; couch.fergus@mayo.edu; 16Cancer Genetics Program, Beaumont Hospital, Royal Oak, MI 48073, USA; dana.zakalik@corewellhealth.org; 17Department of Medicine and Surgery, University of Milan Bicocca, 20126 Monza, Italy; robert.fruscio@unimib.it; 18Beth Israel Deaconess Medical Center, Boston, MA 02215, USA; ntung@bidmc.harvard.edu; 19Moffitt Cancer Centre, Tampa, FL 33612, USA; alvaro.monteiro@moffitt.org; 20Institute of Medical Science, Faculty of Medicine, University of Toronto, Toronto, ON M5S 1A1, Canada

**Keywords:** *BRCA* variant, breast cancer, exon, nucleotide position, risk

## Abstract

Women who carry a pathogenic variant (mutation) in the BRCA1 gene face a high lifetime risk of developing breast (and ovarian) cancer. We evaluated whether the specific type of mutation or the location of the mutation on the gene impacts the risk of developing breast cancer among 3677 BRCA1 carriers. After following these women for an average of 7.2 years, 481 (13.1%) developed breast cancer. Overall, we observed that there was no significant difference in a woman’s risk based on the specific mutation. We did, however, observe a lower risk among women who carried a mutation commonly observed in certain populations or ethnicities; however, this difference would not impact upon clinical management. Additional studies are necessary before personal cancer risk prediction can include specific type and/or location of the BRCA1 mutation.

## 1. Introduction

The breast cancer susceptibility gene 1 (*BRCA1*, *NM_007294.4*) contains 23 exons and comprises 5592 nucleotides, encoding an 1863-amino-acid protein involved in numerous tumour suppressive functions, notably, DNA damage repair by homologous recombination [1]. Loss of *BRCA1* function results in genomic instability, increasing the risk of oncogenic transformation [2]. Approximately 10% of breast cancer cases are hereditary, and germline pathogenic variants (PVs) in *BRCA1* account for nearly 35% of these [3]. PVs in the *BRCA1* gene confer the highest known lifetime risk of breast cancer, estimated to be 72% by age 80 [4,5]. Several thousand different PVs have been reported in *BRCA1* genetic sequence databases [6].

There has been much interest in estimating cancer penetrance according to the specific type and location of a mutation within the *BRCA1*. In 2002, Thompson et al., reported that mutations within the central region of the gene (nucleotides 2401–4190) were associated with significantly lower breast cancer risk than other mutations, while conferring higher risks of ovarian cancer [7]. This region largely overlaps with the ovarian cancer cluster region (OCCR) (nucleotides 1380–4062), later defined by Rebbeck et al., [8]. The OCCR is situated within, and adjacent to, exon 11 of *BRCA1* (hereafter referred to as *exon 10*, according to the NM_007294.4 transcript). Mutations in the OCCR have been associated with relatively high risks of ovarian cancer. In contrast, mutations located in the 5′ and 3′ regions of the *BRCA1* gene have been associated with relatively higher risks of breast cancer, and these regions are called the breast cancer cluster regions (BCCRs). These estimates are based on one large association study of affected and unaffected women, potentially introducing selection bias by including women diagnosed with cancer prior to study enrollment [8].

The risk of breast cancer in *BRCA1* PV carriers may vary by age, family history, and reproductive history [5,9,10]. Currently, risk assessment tools used do not differentiate cancer risk according to the specific *BRCA1* mutation (except for the lower penetrance 5096G > A missense mutation) [11]. If there is substantial variation by PV type and location, this knowledge may help personalize risk-reduction and surveillance approaches. In some countries (e.g., Poland, Iceland), the majority of mutation carriers are represented by a small number of “founder” mutations [12]. These are genetic changes that appear in the DNA of one or more individuals who are the descendants of “founders” in the population and propagated through subsequent generations. It is also of interest to know if founder mutations confer variant-specific cancer risks. Here, in a prospective cohort of unaffected women with a *BRCA1* PV, under imaging surveillance, we seek to determine the impact of PV position and function on cancer risk.

## 2. Materials and Methods

### 2.1. Study Population

We included women with a pathogenic (or likely pathogenic) germline *BRCA1* variant (*carriers* hereafter) enrolled in a prospective, longitudinal, multicenter study. Genetic testing was conducted based on a suggestive family history or a known family PV, but testing criteria varied across the participating centers. PV detection was performed using a range of techniques, with all nucleotide sequences confirmed by direct DNA sequencing. This study was approved by the Institutional Review Board of each participating center, and all participants provided written informed consent.

### 2.2. Data Collection

Each participant completed a baseline questionnaire at the time of study enrollment and a biennial follow-up questionnaire thereafter to update information on a variety of exposures and screening practices and to record incident disease and treatments received. Breast cancer diagnoses, based on self-report (by the patient or next-of-kin), included (1) invasive cancer; (2) ductal carcinoma in situ (DCIS), and (3) mixed cases with both invasive and in situ components. In the event of a cancer diagnosis, pathology reports or medical records were requested to determine the primary site and histology and were available for 75% of the cases in the current analysis.

### 2.3. Subjects Available for Analysis

A total of 13,022 *BRCA1* carriers were enrolled in the parent study between January 1996 and May 2025 (Appendix A). Of these, 9345 subjects were excluded due to the following reasons: personal history of (invasive and/or in situ) breast cancer prior to the baseline questionnaire (*n* = 5442), personal history of ovarian or fallopian tube cancer prior to the baseline questionnaire (*n* = 1400), no follow-up data (*n* = 1699), preventive bilateral mastectomy before enrollment (*n* = 221), or missing data on key variables (*n* = 583). After these exclusions, 3677 *BRCA1* carriers with no prior diagnosis of breast or ovarian/fallopian tube cancer were included in the final analysis.

### 2.4. Variant Classification

At the time of study enrollment, participants were asked to provide a copy of their genetic testing report, or the specific PV was directly provided by the recruiting center. All variants were classified according to the American College of Medical Genetics (ACMG) guideline [13]. Only ‘pathogenic’ or ‘likely pathogenic variants’ were included. Assigned classifications were compatible with *BRCA* Exchange and ClinVar for the variants reported in these databases [6,14]. PVs were grouped, based on type and location within *BRCA1* according to the following criteria:

**(1) Exon**. The analysis was performed according to each individual exon, based on the NM_007294.4 transcript. Due to its large size, exon 10 was further subdivided into four regions of equal size based on the NM_007294.4 transcript, (regions 10.1, 10.2, 10.3 and 10.4) and used region 10.4 as the reference because it included the largest number of PV carriers (n = 342). We also compared risk of cancer in those with a PV in exon 10 vs. outside this exon (“other” PV group).

**(2) Type of variant**. *BRCA1* variants were classified based on the impact on the gene sequence and resultant protein (i.e., frameshift insertion, frameshift deletion, initial codon, missense, nonsense, splicing and large rearrangement). For the breast cancer risk analysis, the different PV types were compared using frameshift deletions as the reference group.

**(3) Breast cancer cluster regions (BCCRs)**. Based on the classification proposed by Rebbeck et al., BCCR1 was defined as the region spanning from c.179 to c.505. BCCR2 included two distinct segments: c.4328–c.4945 and c.5261–c.5563. Risk of developing breast cancer for PVs in each BCCR were compared with PVs outside of the two BCCRs [8].

**(4) Nonsense Mediated Decay (NMD)**: NMD of mRNA was considered to be present for truncating variants located between c.150 and c.5412 according to NM_007294.4 transcript.

**(5) Founder mutations**. We analyzed four known Polish and Jewish founder variants with large number of carriers (100 and more). For each founder variant, the reference group for breast cancer risk analysis was all other PVs.

### 2.5. Statistical Analysis

The study estimated the incidence and 15-year cumulative risk of breast cancer among BRCA1 carriers, according to the type and location of the mutation. Patients were followed from the date of study enrollment (i.e., baseline questionnaire) until the first of either: (1) date of a breast cancer diagnosis (invasive or DCIS); (2) date of risk-reducing bilateral mastectomy; (3) date of completion of the last follow-up questionnaire or (4) date of death from any cause. The cumulative incidence of breast cancer (any) by mutation classification was estimated using the Kaplan–Meier method. Hazard ratios (HR) and 95% confidence intervals (CI) were estimated using the Cox proportional hazards regression adjusting for age. *p*-values were based on two-sided tests and considered to be significant at the level of <0.05. All analyses were performed using SAS Proprietary Software 9.4 (SAS Institute Inc., Cary, NC, USA).

## 3. Results

### 3.1. Study Population

A total of 3677 unaffected *BRCA1* carriers were included in the study. The mean age of study entry was 38.3 years (range 18.0–88.4). The mean follow-up time was 7.2 years (range 0.1–15.0). The distribution of PVs by exon is presented in Appendix A. Among these, 1476 (40.1%) PVs were in exon 19 (including 1437 carriers of the c.5263_5264insC founder mutation), 935 (25.4%) were in exon 10 (the largest exon of *BRCA1*) and 1266 PVs (34.4%) were in other exons (no carriers had PVs in exons 1, 8, or 9).

There were 481 incident cases of breast cancer identified in the follow-up period. Medical records were available for 75% of the cases. The mean age at diagnosis was 46.2 years (range 25.5–86.3). Overall, the cumulative risk of cancer was 8.4% at five years, 17.2% at ten years, and 24.8% at fifteen years. The annual risk of breast cancer, from age 30 to 75, was 2.1%.

### 3.2. Risk of Breast Cancer by Exon

The 15-year cumulative incidence of breast cancer ranged from 9% in exon 21 to 57% for carriers with PVs in exon 6 but many estimates were based on small strata of less than 20 carriers (Appendix A, Figure 1). Compared to exon 12, exon 16 PVs showed a significantly increased risk, although this was based on eight cases (HR = 2.90; 1.20–7.01; *p* = 0.02). Among the 935 carriers of known specific PVs in exon 10, 13.7% developed breast cancer (n = 128 cases) with the highest proportion of cases seen with PVs located in the 10.3 region (Table 1). PVs involving this specific region of exon 10 (10.3) were associated with a 15-year cumulative risk of 37.1% compared with 24.3% in the other exons or 19.2 for PVs in exon 12 (Table 1, Appendix A). The associated HR was 1.59 (95% CI 1.10–2.29).

### 3.3. Risk of Breast Cancer by Specific Mutation Type

Most variants were frameshifts. There were 1620 insertion (44.1%) and 996 deletion (27.1%) frameshift variants, 573 (15.6%) missense, 326 (8.9%) nonsense and 80 (2.2%) large rearrangement variants. Less frequent PVs included: splicing (*n* = 42; 1.1%); intronic (*n* = 29; 0.8%), initial codon (*n* = 7; 0.2%), and in-frame deletions (*n* = 4; 0.1%). There was no significant difference in the 15-year cumulative risk of developing breast cancer according to type of *BRCA1* mutation (*p* = 0.80) (Figure 2). There was no difference in penetrance estimates in the analysis that evaluated the individual impact of the missense founder mutation c.181T > G (p.C61G) previously suggested to be of higher penetrance (*n* = 536) and the other missense variants (*n* = 37) -.

In the analysis stratified according to NMD status, the 15-year cumulative risk was 21.8% for non-NMD mutations and 25.2% for NMD mutations carriers (*p* = 0.43) (Appendix A). Similarly, when comparing missense PVs to protein-truncating PVs, no significant differences were observed in the 15-year cumulative risk, with rates of 24.1% and 24.9%, respectively (*p* = 0.55) (Appendix A).

### 3.4. Cluster Regions and Breast Cancer Risk

During the follow-up period, 280 (60.2%) of the breast cancer cases occurred in carriers with PVs located within the two BCCRs. The 15-year cumulative risks of breast cancer in BCCR1 and BCCR2 were 23.4% and 24.5%, respectively. The corresponding risk estimates in the analysis excluding the four founder mutations were 31.7% and 25.6%, respectively. These values were similar to the risk observed in carriers with mutations outside of these cluster regions (25.4%; *p* = 0.58) (Appendix A).

### 3.5. Risk of Breast Cancer for Carriers of Founder Mutations

Four unique PVs in *BRCA1* were observed more than 100 times each and accounted for 66% of the carriers in this analysis: (1) c.4034delA (145 times), (2) c.66_67delAG (315 times), (3) c.181T > G (536 times), and (4) c.5263_5264insC (1437 times). The 15-year cumulative risk of breast cancer associated with these PVs ranged from 15.9% (c.4034delA) to 24.4% (c.5263_5264insC). In comparison, carriers of less frequent PVs, grouped together under the ‘*other*’ category, exhibited a cumulative risk of 28.8% (Table 2, Figure 3). When compared to “other” PVs group, all founder mutations were associated with a lower risk of disease; however, only the association with c.4034delA reached statistical significance after adjusting for age (HR = 0.49; 95% CI 0.27–0.89; *p* = 0.02) (Table 2).

## 4. Discussion

In this prospective cohort of 3677 unaffected *BRCA1* carriers, the annual breast cancer risk from age 30 to 75 was 2.1%, with a 15-year cumulative risk of 24.8%. There was some variation by specific exon (lower in exon 12~19% and higher for exon 16~50%); however, strata were small in some instances and no statistically significant differences emerged. Two-thirds of carriers in this analysis harbored a PV in one of four founder mutations, with the lowest risk observed for c.4034delA founder (15.9%). The distribution of breast cancer cases appeared to be similar for PVs within and outside of BCCRs. In a previous analysis of affected and unaffected *BRCA1* carriers (n = 19,581), Rebbeck et al., reported that the relative risk of developing breast cancer was not uniformly distributed across the *BRCA1* gene but depended on both type and location of the PVs [8]. In a recent analysis of the CARRIERS consortium, which included over 32,000 cases and matched controls, Akamandisa et al., did not observe any differences in the odds of disease by *BRCA1* PV although the number of carriers was low (0.3%) [15]. Our findings suggest that, while some variability exists, *BRCA1*-breast cancer risk may be less strongly influenced by specific PV than previously thought. In contrast to the other reports, our main outcome was cumulative incidence rather than relative risk, which is more practical in terms of communicating risk to carriers during their post-test consultation.

In our study, some comparisons reached statistical significance, but a large number of comparisons were presented. Notably, the 15-year cumulative breast cancer risk was lowest for exon 12 (19%) and highest for exon 16 PVs (50%). Using exon 12 as reference group, there was a borderline significant increased breast cancer risk with PVs in exon 16 (HR = 2.44; 95% CI 1.01–5.88). Exon 16 overlaps codon 1525–1604 and there is no known structural or functional domain in this region. Compared to PVs in exon 12, PVs in exon 10.3 were associated with a significantly increased risk (HR = 1.68; 95% CI 1.17–2.43). Exon 10 (referred to as exon 11 in earlier studies) constitutes approximately 60% of *BRCA1* coding sequence [16]. It harbors critical functional motifs, including two nuclear localization signals required for tumor suppressor activity [16,17,18]. It has been proposed that PVs in exon 10 lead to a decrease in the tumor suppressor activity of *BRCA1,* by reducing DNA repair activity, and consequently to an increase in unrepaired mutations and chromosomal abnormalities [19].

Most patients in our study (66.2%) carried one of four founder mutations, common in Eastern Europe. This is related to our high accrual rate of subjects from Poland (53%) and of Jewish women (9%), most of whom had a founder mutation. The four specific founder mutations in this cohort were all associated with a lower, but still clinically relevant penetrance than the other PVs. The relatively low risk seen among carriers of high-frequency founder mutations may be due to ethnic background (mostly Eastern European), non-genetic risk factors and differences in family history. It may also reflect more widespread testing for founder mutations than non-founder mutations due to relaxed clinical criteria.

The c.4034delA PV is a frameshift deletion located in exon 10, common in the Southern Baltic region. This variant has previously been described as a reduced-penetrance PV for breast cancer [20,21]. This was confirmed in our cohort; the estimated 15-year cumulative breast cancer risk was 15.9%, the lowest observed among the four founder mutations analyzed (HR = 0.49; 95% CI 0.27–0.89).

The c.181T > G variant is a missense variant, located in exon 5 within the RING finger domain of the *BRCA1* protein. It is relatively frequent in Central and Eastern Europe [22,23]. In the Rebbeck et al., study, missense PVs in the RING domain were associated with a significantly increased breast cancer risk (HR = 1.56; 95% CI 1.32–1.84) compared to exon 10 missense PVs [8]. In contrast, in our study, carriers of this missense PVs had a 15-year cumulative risk of 23.1%, with a hazard ratio below 1.0 (HR = 0.83; 95% CI 0.83–1.09) compared to PVs in exon 12.

The c.66_67delAG is a frameshift deletion located in exon 2, involving the RING domain of the *BRCA1* protein. This PV is a founder mutation in the Ashkenazi Jewish population, but it has been observed in other populations as well [20,24]. In our cohort, carriers of this PV had a lower breast cancer incidence compared to the other non-founder PV group (HR = 0.70; 95% CI 0.48–1.03) but the difference was not statistically significant. Others have shown no difference in risk estimates in carriers of this founder variant [25,26].

The c.5263_5264insC is a common frameshift insertion involving exon 19, a known Slavic PV with founder effects in Eastern European as well as Ashkenazi Jewish communities [27]. This single mutation represents 94%, 73% and 60% of *BRCA1* mutations reported in Russia, Belarus and Poland, respectively [23]. In a previous study, the lifetime risk of breast cancer in carriers of the c.5263_5264insC PV was estimated to be 67% [28]. In the current cohort, the 15-year cumulative risk was 24.4%. The risk was slightly lower than that of all other non-founder PVs (HR = 0.86; 95% CI, 0.70–1.06) and contrasts findings from the Rebbeck et al., analysis reporting a HR of 1.63 (95% CI 1.41–1.89) when including exon 10 nonsense mutations as the comparator group [8].

We did not observe significant differences in breast cancer risk for PVs located within vs. outside the BCCRs, in contrast to findings of Rebbeck et al., [8]. This difference may partly reflect the overlap of BCCR1 with the founder PV c.181T > G and BCCR2 with c.5263_5264insC, both relatively frequent in our population and associated with lower breast cancer risks. The presence of these founder PVs may explain the lack of observed risk differences between BCCR and non-BCCRs. Furthermore, differences from the previous study may be influenced by variations in population characteristics and study design.

There are several limitations to our study. Although we included 3677 women, many of the strata were small and confidence limits wide. Women were followed for an average of 7.2 years, and additional follow-up will continue to help refine these risks. Given that a large proportion of our study population carried a founder PV (66.2% of the total), many of the overall risk estimates were driven by these four PVs, potentially limiting generalizability of our findings. We could not evaluate the impact of the lower risk variant c.5096G > A (p.R1699Q), as only eight carriers were identified [11]. Differences between our results and others may reflect variations in study design. For example, Rebbeck et al., in an observational study, included prevalent cases and initiated follow-up at birth in women with a known mutation, while Akamandisa et al., included population-based cohorts with 308 *BRCA1* PV carriers (273 cases and 35 controls), and estimated odds of disease [8,15]. Our prospective cohort included exclusively unaffected women, thereby minimizing population selection bias related to testing women with a prior diagnosis of cancer, but may have introduced survivorship bias. Finally, although age-adjusted risks were reported, we did not include other covariates given that these are unlikely to correlate with specific PV.

## 5. Conclusions

In summary, we conclude that although risk of developing breast cancer is different for specific PVs, it is premature to use type and location of PVs to stratify *BRCA1* carriers about their personal cancer risk. Recommendations regarding preventive surgery and MRI surveillance should not be solely variant-dependent and should integrate other factors such as personal and family history of cancer as well as ages at diagnosis [29,30,31]. Compared to non-founder PVs, carriers of one of four founder PVs experienced a lower breast cancer risk. The variation in risk for carriers of PVs based on location (e.g., higher for exons 6, 10.3 and 16; lower for exon 12 and 21) requires further validation. Follow-up of this and other cohorts are necessary to confirm our findings and continue to refine cancer risks.

## Figures and Tables

**Figure 1 curroncol-32-00705-f001:**
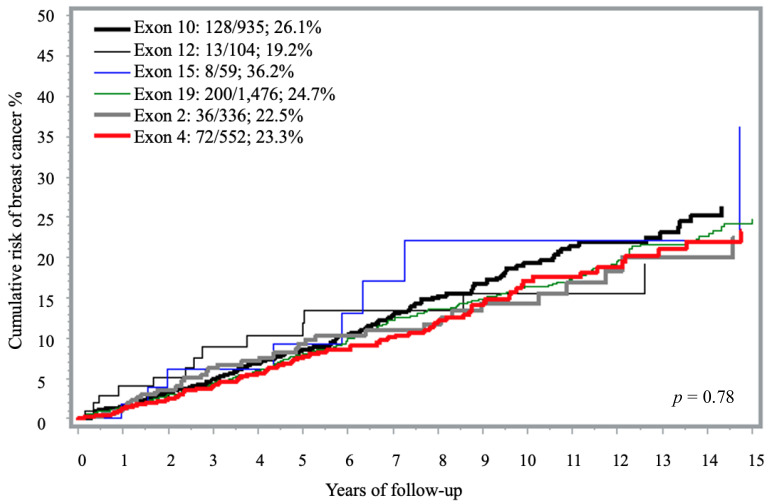
Fifteen-year cumulative risk of breast cancer among *BRCA1* carriers, by exon location ^a^. ^a^ The analysis included only exons represented by more than 50 carriers. Exons with fewer than 50 carriers were excluded due to limited sample size (i.e., exons 1, 3, 5–9, 11, 13–14, 16–18 and 20–23).

**Figure 2 curroncol-32-00705-f002:**
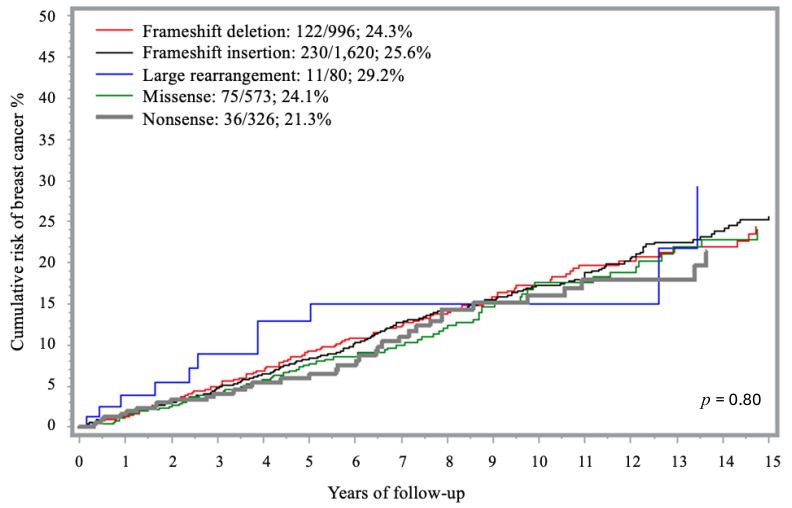
Fifteen-year cumulative risk of breast cancer among *BRCA1* carriers, by PV type.

**Figure 3 curroncol-32-00705-f003:**
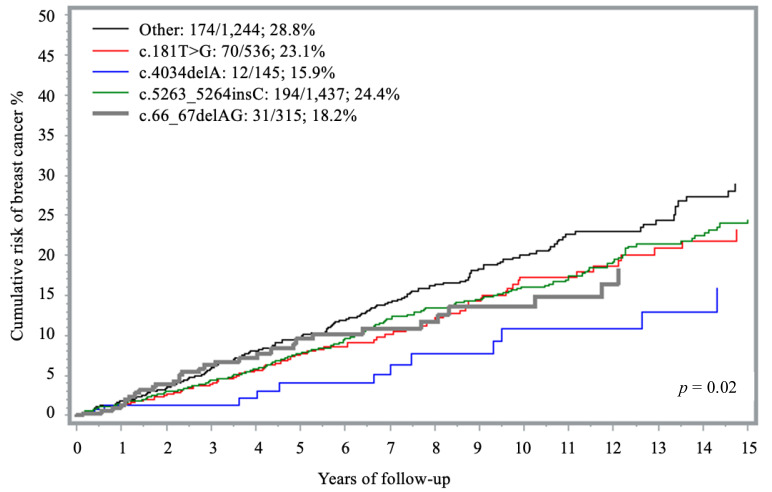
Fifteen-year cumulative risk of breast cancer among *BRCA1* carriers, by founder mutation.

**Table 1 curroncol-32-00705-t001:** Fifteen-year cumulative risk of breast cancer among *BRCA1* carriers, by exon 10 regions.

Exon 10 Regions	n (%)	Breast Cancer Events, n (%)	15-Year Cumulative Risk	Exon 10.1, 10.2, 10.3 vs. 10.4				Exon 10.1, 10.2, 10.3, 10.4 vs. Other			
				HR (95% CI) ^a^	*p*	HR (95% CI) ^b^	*p*	HR (95% CI) ^c^	*p*	HR (95% CI) ^b^	*p*
10.1	251/3677 (6.8)	38/251 (15.1)	34.8%	1.51 (0.96–2.38)	0.07	1.46 (0.93–2.30)	0.10	1.33 (0.95–1.86)	0.09	1.25 (0.89–1.75)	0.20
10.2	176/3677 (4.8)	21/176 (11.9)	19.9%	1.03 (0.61–1.76)	0.91	0.99 (0.58–1.69)	0.98	0.91 (0.58–1.41)	0.65	0.85 (0.55–1.32)	0.46
10.3	166/3677 (4.5)	31/166 (18.7)	37.1%	1.92 (1.19–3.09)	0.007	1.86 (1.15–2.99)	0.01	1.68 (1.17–2.43)	0.005	1.59 (1.10–2.29)	0.01
10.4	342/3677 (9.3)	38/342 (11.1)	20.4%	1.00 (ref.)		1.00 (ref.)		0.87 (0.62–1.21)	0.40	0.85 (0.61–1.19)	0.35
Total exon 10	935/3677 (25.4)	128/935 (13.7)	26.1%					1.12 (0.92–1.37)	0.26	1.07 (0.87–1.31)	0.51
Other	2742/3677 (74.6)	353/2742 (12.9)	24.3%	n/a				1.00 (ref.)		1.00 (ref.)	
**All exons**	**3677 (100)**	**481/3677 (13.1)**									

Breast cancer events and 15-year cumulative risk according to *BRCA1* exon 10 regions (10.1, 10.2, 10.3, and 10.4). Hazard ratios (HRs) were estimated for individual exon 10 regions using exon 10.4 as reference, given its highest number of carriers. Additionally, HRs were analyzed for individual exon 10 regions and for exon 10 overall compared with PVs outside exon 10 (reference group). All HRs were also adjusted for age. Abbreviations: HR, Hazard Ratio; CI, confidence interval; ref., reference; n/a: not applicable. ^a^ Exon 10.4 used as reference group. ^b^ Age-adjusted. ^c^ “Others” includes mutations outside of exon 10 as reference group.

**Table 2 curroncol-32-00705-t002:** Fifteen-year cumulative risk of breast cancer among *BRCA1* carriers, by founder mutation.

Variant	n (%)	Breast Cancer Events, n (%)	15-Year Cumulative Risk	HR (95% CI)	*p*	HR (95% CI) ^a^	*p*
c.4034delA	145/3677 (3.9)	12/145 (8.3)	15.9%	0.45 (0.25–0.81)	0.01	0.49 (0.27–0.89)	0.02
c.66_67delAG	315/3677 (8.6)	31/315 (9.8)	18.2%	0.74 (0.51–1.09)	0.12	0.70 (0.48–1.03)	0.07
c.181T > G	536/3677 (14.6)	70/536 (13.1)	23.1%	0.76 (0.58–1.01)	0.06	0.83 (0.62–1.09)	0.18
c.5263_5264insC	1437/3677 (39.1)	194/1437 (13.5)	24.4%	0.80 (0.65–0.98)	0.03	0.86 (0.70–1.06)	0.15
Other ^b^	1244/3677 (33.8)	174/1244 (14.0)	28.8%	1.00 (ref.)		1.00 (ref.)	
**Total**	**3677 (100)**	**481/3677 (13.1)**					

Abbreviations: HR, Hazard Ratio; CI, confidence interval; ref., reference. ^a^ Age-adjusted. ^b^ “Others” includes all other mutations as reference group.

## Data Availability

Data are not publicly available. All data are available through the Risk Factor Study repository upon application. The study was conducted in accordance with the Declaration of Helsinki, and approved by the Institutional Review Board of Women’s College Hospital (This study was conducted under protocol number 2007-0036-B.).

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
