# Peer review of "The Risk of Breast Cancer According to Mutation Type and Position in Carriers of a Pathogenic Variant in BRCA1"

_curroncol, 2025, doi:10.3390/curroncol32120705_

Round 1

Reviewer 1 Report

Comments and Suggestions for Authors

The study addresses an important issue in the relationship between the type and position of BRCA1 pathogenic mutation and breast cancer risk. The study identified 481 breast cancer by following 3,677 BRCA1 PV carriers for an average of 7.2 years. Through extensive comparison of the cancer patients, the study observed no significant differences for breast cancer risk between mutation type, mutation position. The study concludes that it is premature to use mutation type and location information to guide clinical practices for BRCA1 mutation carriers. However, this conclusion may be influenced by multiple conditions used in this study.

Comments

  1. Exon-based risk estimates. The size of 23 exons in BRCA1 is very different, with exon10 accounts for 60% of BRCA1 coding aa. Although the study divided exon10 into 4 subsections, it is still not a fair comparison between different exons. Beyond the comparison between exons, it would be interesting to see if the conclusion could be changed by comparing the normalized regions based on the same number of aa residues across the entire BRCA1 coding region;
  2. Founder mutations. The study used the frequency of PV carriers (100 times) to define founder PVs and non-founder PVs. This definition is unacceptable, as the frequency can be influenced by multiple factors, such as the different portion of different PV carriers. For example, c.66_67delAG and c.5263_5264insC are the founder mutation commonly present in Ashkenazi Jews. Their determination as founder mutation is based the evidence from haplotyping analysis, not by frequency. The size of Ashkenazi Jews population is minor comparing to global human population. Further, the four “founder mutation” carriers accounted for 66% of the mutation carriers in the cohort. The insignificant observation could be largely influenced by the four PVs over other PVs for cancer risk estimates;
  3. The study challenges the claimed BCCR region. This raised an interesting question for the reliability of BCCR. Further explore this aspect would be worth, at least in the Discussion part;
  4. Exon 16 PVs had the highest risk. As there is no known functional element in exon 16, the structural importance of exon 16 may give certain clues to explain the high risk for exon 16 PVs;
  5. BC risk increases by age, especially after reproduction period. The observed lower penetrance in founder mutation carriers may be confounded by systematic age differences at enrollment and differential follow-up time. Younger age at enrollment translates to shorter follow-up time in the cancer-free state before reaching ages of peak breast cancer incidence (50-70 years). Comparing different age groups may be needed to see if significance could be present;
  6. Multiple typos: BRCA1 carriers should be BRCA1 PV carriers; BRCA1 should be italic if referring to gene and non-italic if referring to protein; specific BRCA1 mutation should be specific BRCA1 variant as per ACMG/AMP nomenclature standards that “variant” is preferred in general contexts and mutation should be used only for established terms such as “founder mutation”.

Reviewer 2 Report

Comments and Suggestions for Authors

The authors present a large prospective cohort study of 3,677 unaffected BRCA1 pathogenic variant (PV) carriers to evaluate whether breast cancer risk varies by mutation type and genomic position. The topic is clinically relevant, particularly in the context of precision risk stratification and genetic counseling. The manuscript is clearly written, generally well organized, and addresses an important clinical question regarding whether BRCA1 mutation location should influence individualized cancer risk management.

The dataset is large and prospectively collected, which strengthens the validity of the conclusions. However, there are several issues that should be addressed before the manuscript is suitable for publication.

Strengths of the Study

  1. Largest prospective cohort restricted to unaffected BRCA1 carriers.
  2. Avoids retrospective survivor bias common in earlier studies.
  3. Addresses a relevant and debated clinical question.
  4. Good integration with previous literature (Rebbeck, CARRIERS, etc.).
  5. Results will be useful for future meta-analyses and PV-stratified counseling tools.

Major Comments

  1. The study acknowledges known modifiers of breast cancer risk (e.g., parity, breastfeeding, hormonal contraception, BMI, prophylactic oophorectomy), but none were included in adjusted models.
    → Recommendation: Provide rationale for excluding these variables or adjust for them where available.
  2. Approximately 66% of participants carry one of four founder mutations. This creates a population structural imbalance, limiting generalizability.
    → Clarify whether findings apply primarily to Eastern European populations and how this affects risk interpretation for diverse populations.

Minor Comments

  1. The definition of “founder mutation” (>100 carriers) is arbitrary; justify threshold or cite precedent.
  2. In Introduction, briefly update current clinical guidelines regarding PV-specific risk assessment (NCCN 2025 and ASCO 2024 are cited, but not summarized).
  3. Minor typo: “risk of breast cancer in BRCA1 PV carrier may vary…” → “carriers”

Reviewer 3 Report

Comments and Suggestions for Authors

This manuscript by Kotsopoulos et al. presents a valuable analysis from a large, prospective, multicenter cohort study investigating whether the type and location of pathogenic variants (PVs) in the BRCA1 gene influence breast cancer risk. The study includes 3,677 unaffected BRCA1 carriers followed for a mean of 7.2 years. The primary finding is that while some variability in 15-year cumulative breast cancer risk was observed by exon and for specific founder mutations, the overall differences were modest and often based on small sample sizes. The authors conclude that it is premature to use PV-specific information for personalized risk stratification in clinical practice. The study is well-designed and addresses an important clinical question, but several aspects require clarification and the limitations should be more thoroughly discussed.

Concerns:

  1. The choice of reference groups in different analyses needs clarification. For example, in Table 1, the risk for exon 10.1-10.3 are presented using exon 10.4 as a reference and also using "all other exons" as a reference. The rationale for these different comparisons should be explained.
  2. In the founder mutation analysis, the "Other" group is the reference. It would be helpful to know the composition of this "Other" group. Does it contain a diverse mix of variants, or is it also dominated by a few common non-founder PVs?

  3. The discussion contrasts the findings with those of Rebbeck et al. (2015) and Akamandisa et al. (2025), suggesting that PV-specific risk may be less influential than previously thought. This is a key argument. This point could be strengthened by more directly discussing potential reasons for the discrepancies beyond study design (prospective vs. case-control). 

  4. The caption for Figure 1 states that exons with fewer than 50 carriers were excluded. It would be useful to state in the caption or the main text how many exons were excluded for this reason.
  5. The manuscript states that cancer diagnoses were based on self-report, with subsequent attempts to obtain pathology reports. For transparency, please report the percentage of self-reported cancer cases for which pathological confirmation was successfully obtained.
  6. The mean follow-up of 7.2 years is reasonable, but the range of 0.0-15.0 years suggests some participants contributed very little time. A sensitivity analysis excluding participants with very short follow-up (e.g., <1 year) could be considered to ensure they are not unduly influencing the estimates.

Round 2

Reviewer 1 Report

Comments and Suggestions for Authors

The revision addressed each of my questions satisfactorily.

Reviewer 3 Report

Comments and Suggestions for Authors

The authors addressed all my concerns. Recommend for acceptance.